# A single *Musashi* gene allele is sufficient to maintain mouse photoreceptor cells

Bohye Jeong, Peter Stoilov *

Department of Biochemistry and Molecular Medicine, West Virginia University (WVU), Morgantown, West Virginia, United States of America

* pstoilov@hsc.wvu.edu

## Abstract

In vertebrates, two genes, *Musashi1 (Msi1)* and *Musashi2 (Msi2)*, encode for highly similar *Musashi* protein paralogs. The *Musashi* proteins are known to bind to 3'-UTRs and control translation. In photoreceptor cells, the *Musashi* proteins promote the inclusion of photoreceptor-specific alternative exons by binding to the proximal downstream of their introns. While the *Musashi* proteins are expressed in various cell types, their role in regulating splicing appears to be confined to photoreceptor cells, where the two proteins have exceptionally high expression levels. To test if the photoreceptor-specific role of MSI1 and MSI2 in splicing is due to their expression levels in photoreceptor cells, we generated combined *Msi1* and *Msi2* knockouts that progressively reduced the number of *Musashi* alleles in photoreceptor cells. We analyzed the splicing of photoreceptor-specific exons in the *Cc2d2a*, *Cep290*, *Prom1*, and *Ttc8* genes and the function of photoreceptor cells in the knockouts. We found that a single allele from either *Msi1* or *Msi2* is sufficient to maintain photoreceptor function and support high inclusion levels of the photoreceptor-specific exons.

## Introduction

The *Musashi* proteins are a family of RNA binding proteins conserved in metazoans. First discovered in *Drosophila, Musashi (dMsi)* was implicated in the maintenance of neural and germ stem cells [1–4]. *The Drosophila genome* contains *a paralog dMsi*, *Rbp6,* but the two proteins appear to perform nonredundant functions in the fly [5]. Vertebrate genomes contain two *Musashi* paralogs, *Msi1* and *Msi2,* that are more closely related to *Rbp6* than *dMsi* [5]. The vertebrate *Musashi* genes were shown to support multiple stem cell populations and stem cell fate transition in mammals, and control the expression of maternally produced transcripts in *Xenopus oocytes* [6–10].

The canonical role of the *Musashi* proteins is to bind to the 3'-UTRs of their mRNA targets to regulate translation [11–13]. Translation is regulated by the Musashi proteins through distinct mechanisms that involve interactions with the cytoplasmic poly-A polymerase, poly-A binding proteins, and LSM14B [14–17].

**Data availability statement:** Supplementary data set containing raw gel and microscopy images (doi:10.7910/DVN/TBWANU) is available at Harvard Dataverse under following URL: https://doi.org/10.7910/DVN/TBWANU All other relevant data are within the paper and its Supporting Information files.

**Funding:** R01EY025536. The funders had no role in study design, data collection and analysis, decision to publish, or preparation of the manuscript.

**Competing interests:** The authors have declared that no competing interests exist.

In vertebrates, MSI1 and MSI2 are expressed across multiple cell types, including terminally differentiated neurons. However, their functions outside of stem cell populations are less well understood. Recently, the *Musashi* proteins were shown to regulate the plasticity of differentiated gonadotrope cells in the pituitary by controlling protein translation [18]. We have previously demonstrated that MSI1 and MSI2 are particularly abundant in the vertebrate retina, where they are required for the development and maintenance of photoreceptor cells [19,20]. In photoreceptor cells, the *Musashi* proteins promote the expression of a broad range of proteins [19]. Unique to photoreceptor cells, the *Musashi* proteins promote the inclusion of alternative exons when bound to the downstream proximal intron [21].

We were interested in understanding why the *Musashi* proteins play such a prominent role in regulating alternative splicing specifically in photoreceptor cells, despite being expressed in many mammalian cell types. The binding of MSI1 to RNA has been shown to inhibit its translocation to the nucleus, as its nuclear localization signal (NLS) is also part of the RNA-binding surface [22]. Thus, we reasoned that the stoichiometry between *Musashi* proteins and their cytoplasmic RNA targets may dictate the amount of *Musashi* protein in the nucleus. High levels of Musashi proteins can saturate their cytoplasmic mRNA targets and the excess free protein will be imported into the nucleus to regulate splicing. Such a mechanism will be consistent with exceptionally high expression of *Musashi* proteins in the retina and the activation of photoreceptor-specific exons in cultured cells by overexpression of MSI1 [20,21].

To test this hypothesis, we manipulated the *Musashi* gene dosage in mature photoreceptor cells by generating a series of *Msi1* and *Msi2* knockouts that eliminated one, two, three, or all four of the combined *Msi1* and *Msi2* alleles. We observed a complete redundancy between *Msi1* and *Msi2* in photoreceptor cells, where a single allele from either gene was sufficient to support the splicing of photoreceptor-specific exons and photoreceptor function.

## Materials and methods

### Animal care, anesthesia, and euthanasia

Animal experiments in this study were carried out with the approval of the Institutional Animal Care and Use Committee (IACUC) at West Virginia University (WVU).

Animal care: All mice were housed in the pathogen-free animal facility with a 12 hour light/dark cycle, and *ad libitum* food and water were provided. All mice were monitored daily by trained technicians and state-certified staff veterinarians. Animals were evaluated for signs of pain and distress: eyelids partially closed; hair loss; teeth alignment for malocclusions; increased aggression toward cage mates; reduced exploratory behavior; and aggressive vocalization.

Anesthesia for electroretinography (ERG) procedures: The mice were anesthetized with isoflurane (5%) in oxygen (2.5%) in the induction chamber for 5 minutes. After that the mice were placed onto a heated platform (37 °C) to keep body temperature and the anesthesia was maintained equipped with a nose cone with a constant flow of isoflurane (1.5%) in oxygen (2.5%) through an equipped nose cone throughout the ERG recording.

Euthanasia: $CO_2$ inhalation was used for euthanasia in this study following the guidelines by the Panel on Euthanasia of the American Veterinary Medical Association and approval by WVU IACUC. Animals were euthanized by $CO_2$ inhalation (3%) in an approved delivery chamber, immediately followed by cervical dislocation.

## Animal maintenance and genotyping

Animal lines are routinely backcrossed to C57BL/6J (Jackson Laboratory, Strain # 000664) to avoid the effects of inbreeding. Our animal lines are routinely outcrossed to avoid effects of inbreeding and genotyped for *rd1* and *rd8* mutations that are known to affect vision in mice. The mice were genotyped at weaning using primers listed in S1A Table in S1 File. For *Msi1* genotyping PCR reactions were supplemented with Betain (1M) and DMSO (5%) to minimize mispriming associated with the GC-rich template. Experimental animals were produced by crossing male mice carrying floxed alleles for either or both *Msi1* and *Msi2*, with *Pde6g*<sup>CreERT2</sup> with female mice carrying only floxed alleles. The floxed *Msi1 and Msi2* alleles and the *Pde6g*<sup>CreERT2</sup> mice were described previously [19,23,24]. All the experiments in this study were performed in both male and female mice under C57BL/6J background.

## Tamoxifen-induced knockouts

Three intraperitoneal injections of tamoxifen (Sigma-Aldrich, Catalog # T5648-1G) were conducted on the animals on consecutive days starting at postnatal day 30, as we have done previously [19]. Days post tamoxifen injection are counted from the day of the first injection in the series. To prepare for injections, tamoxifen was thoroughly dissolved in ethanol (Sigma-Aldrich, Catalog # E7023) (100 mg/ml) using a thermal mixer (ThermoFischer, Catalog# 13687717) at 40 °C. After that, it was diluted in corn oil (10 mg/ml). Following vacuum centrifugation at 30 °C for 10 minutes to remove the ethanol, before the tamoxifen (100 mg/kg) is administered via intraperitoneal injection.

## Western blot

Mouse retina were collected 14 days after the first tamoxifen injection and lysed with RIPA buffer (50 mM Tris HCl-pH 8.0, 150 mM NaCl, 1.0% TritonX-100, 0.5% sodium deoxycholate, 0.1% sodium dodecyl sulfate) and protease inhibitor cocktail (Sigma-Aldrich, Catalog# P8340). The protein concentration was measured using a BCA protein assay kit (ThermoFischer, Catalog# 23225) and Synergy H4 hybrid reader (BioTek). Protein samples (30 μg) were resolved in 4–20% gradient polyacrylamide SDS–PAGE gel electrophoresis and transferred to polyvinylidene difluoride (PVDF) transfer membrane (ThermoFischer, Catalog# 88518). Membranes were blocked with BSA (5%) in 1X TBST (10 mM Tris HCl-pH7.5, 150 mM NaCl, 5 mM EDTA, and 0.1% Tween) for 1 hour at room temperature and incubated with primary antibodies diluted in BSA (3%) in 1X TBST overnight at 4 °C. Beta tubulin (TUBB), an abundant cytoskeletal protein in the retina, was used as loading control for consistency in our previous studies on MSI1 and MSI2 [19,20]. The next day, secondary antibodies diluted in BSA (3%) in 1X TBST were used to incubate membranes for 1 hour at room temperature. The membranes were then imaged on Amersham Typhoon (Cytiva). Primary and secondary antibodies used for western blot are listed in S2 Table in S1 File.

## Immunofluorescence

Mouse eye from the animals used in the ERG experiments were enucleated 143 days after the first tamoxifen injection and a small incision was made along the cornea. Following fixation of eyeballs in paraformaldehyde fixative solution (4% PFA in 1X PBS: 137 mM NaCl, 2.7 mM KCl, 10 mM $Na_2HPO_4$, and 1.8 mM $KH_2PO_4$) (Electron Microscopy Sciences, Cat# 15710) for 2 hours at room temperature on a rotator, the eyeballs were dehydrated in sucrose (20% in 1X PBS) (Fischer Scientific, Cat# S5-3) overnight at 4 °C. The next day, the eyeballs were embedded and frozen in optimal cutting temperature (OCT) compound (Sakura, Cat# 4583) in Tissue-Tek cryomold (Sakura, Cat# 4565). The frozen eyeballs

were sectioned with 16 μm thickness using CyoStar NX50 cryostat (Epredia) and then mounted onto Superfrost Plus™ microscope slides (Fischer Scientific, Cat# 1255015). The mounted sections were washed three times with 1X PBS for 5 minutes each and then blocked with a blocking buffer (10% goat sera, 0.5% Triton X-100, and 0.05% sodium azide in 1X PBS) for 1 hour at room temperature. After that, the sections were incubated with primary antibodies diluted in antibody dilution buffer (5% goat sera, 0.5% Triton X-100, and 0.05% sodium azide in 1X PBS) overnight at 4 °C. The following day, the sections were washed three times with 1X PBST (0.1% Triton X-100 in 1X PBS) for 10 minutes each and then incubated with secondary antibodies and DAPI (4',6-Diamidino-2-Phenylindole (ThermoFischer, Cat# 62248) diluted (1:1,000) in antibody dilution buffer for 1 hour at room temperature. After the sections were washed twice with 1X PBST for 15 minutes each and lastly washed once with 1X PBS for 15 minutes, the sections were mounted with ProLong™ Gold Antifade Mountant (ThermoFischer, Cat# P36934) and covered with Microscope Cover Glass (Fischer Scientific, Cat# 12544B). Following securing the coverslip with clear nail polish, the sections were imaged using a Nikon AX inverted confocal microscope. Throughout the scanning process, the imaging settings remained the same for all slides. Primary and secondary antibodies used for immunofluorescence are listed in S2 Table in S1 File.

## Image analysis

Image J was used to quantify the signal intensities across the retinal layers. First the background was subtracted using the rolling ball algorithm with a ball radius of 500 pixels (74 μm). For each retinal section the fluorescence signal profile was collected along the length of three 300 pixel (44 μm) rectangular sections (technical replicates) perpendicular to the retinal layers. Subsequent data analysis was performed in R. The intensity profile was separated into segments corresponding to the inner segment (IS), outer nuclear layer (OS), and the combined inner neuronal layers (Fig 4). The signal intensities for the photoreceptor cells (inner segment and outer nuclear layer) were integrated and normalized to the integrated intensity of the inner neurons in each section). The three technical replicates for each retina were averaged to generate one biological replicate. Two way ANOVA and TukeyHSD tests were used to determine statistical significance. The full set of images used in the analysis is shown on S7 and S8 Figs. Raw data and data analysis code in R are provided in Supplementary data and code 2.

## RNA extraction and RT-PCR

Mouse retina were collected 14 days after the first tamoxifen injection. The RNA was extracted using TRIzol Reagent (ThermoFischer, Catalog# 15596026), chloroform (Fisherscientific, Catalog# C298-500), and isopropanol (Sigma-Aldrich, Catalog # 650447). The RNA was then treated with DNase I recombinant (Sigma-Aldrich, Catalog# 4716728001) for 20 minutes at 37 °C. The reactions were extracted with chloroform and the RNA was precipitated with ethanol and sodium acetate (3 M, pH 5.2). The RNAs (200 ng) were reverse-transcribed to cDNA using random hexamers (50 μM) and oligo-dT (10 μM), which was then amplified with fluorescently labeled primer sets spanning the alternatively spliced region listed in S1B Table in S1 File. The PCR products were resolved on Urea (7.5 M)/ denaturing polyacrylamide gel (4% of 19:1 acrylamide/bis-acrylamide ratio) electrophoresis and scanned on Amersham Typhoon imager (Cytiva). Image Quant software (Cytiva) was utilized to quantify the band intensities on the gels. The individual band intensities and the quantification of the exon inclusion ratio are listed in S3 Table in S1 File.

## Electroretinography (ERG)

Mice were dark-adapted overnight prior to recording ERGs. After that, testing was done under the red light. The mice were anesthetized with isoflurane and placed onto a heated platform (37 °C) equipped with a nose cone with a constant flow of isoflurane. The eyes were dilated using a drop from a 1:1 mixture of tropicamide (1%, Sandoz) and phenylephrine-Hydrochloride (2.5%, Paragon) for 10 minutes. A reference electrode was placed subcutaneously between the ears, and a

ground electrode was placed in the mouse's thigh. The mice eyes were lubricated with GenTeal gel (0.3% Hypromellose, Alcon) before positioning silver wire electrodes near the center of the cornea surface. ERG recordings for both scotopic and photopic responses were collected using UTAS Visual Diagnostic System with UBA-4200 amplifier and interface, Big-Shot Ganzfeld device, and EMWIN 9.0.0. software (LKC Technologies, Gaithersburg, MD, USA). Scotopic ERG was recorded using LED white light flashes at increasing flash intensities (−40, −24, −12, and −4 dB). To record photopic responses, the rod photoreceptors were saturated by light-adapting the mice for 5 minutes using 30 cd-s/m² white background light. After light-adaptation, photopic ERG was recorded using flash intensity (3 dB). ERG data was analyzed in R (Supplementary data and code 1) and the results for scotopic (−12 dB) and photopic (3 dB) stimulation are presented on Fig 3 and S4 Fig

### Statistics analysis

Three replicates were used for western blot and RT-PCR experiments. ERG recordings were performed on four to five animals per group and each eye was treated as a separate replicate in the analysis. Statistical significance was determined by two-way ANOVA. Tukey HSD or pairwise T-test was used for pairwise comparisons as indicated. All data were presented as the mean±standard error of the mean (SEM), unless otherwise noted.

## Results

### Combinatorial ablation of *Musashi 1* (*Msi1)* and/or *Musashi 2* (*Msi2*) alleles in mature photoreceptor cells

To manipulate the total *Musashi* protein levels in mature photoreceptor cells, we crossed mice carrying floxed *Msi1* and/or *Msi2* alleles to *Pde6g^CreERT2* knock-in mice [19]. To account for potential differences in the expression of *Msi1* and *Msi2*, we generated eight combinations of floxed *Msi1* and *Msi2* alleles covering all possible single, double, triple, and quadruple allele knockout combinations (Fig 1A and S1 Fig). To delete the alleles from the mature photoreceptor cells, the animals were injected with tamoxifen at postnatal day 30. Animals with matching floxed *Musashi* alleles but lacking the *Cre* allele were used as controls. We have previously shown that deletion of the *Msi1* or *Msi2* alleles in photoreceptor cells leads to approximately 50–70% reduction of the corresponding protein in retinal lysates [19]. Western blot confirmed the expected reduction in MSI1 and MSI2 protein levels when both alleles for each gene were targeted (Fig 1B and S2 Fig). Western blot samples were collected 14 days after inducing the knockout with tamoxifen, based on a time course we have done previously where Musashi protein levels reached their minimum two weeks after the knockout was induced [19].

### A single allele from either *Msi1* or *Msi2* is sufficient to support splicing of photoreceptor-specific exons

We first sought to understand the effect of *Musashi* gene dosage on alternative splicing. We analyzed the splicing of four *Musashi*-dependent photoreceptor-specific alternative exons in the *Cc2d2a, Cep290, Prom1,* and *Ttc8* genes [20,21]. RT-PCR was conducted on eight knockout allele combinations and matching controls (Fig 2 and S3 Fig). Splicing of the photoreceptor-specific exons in the *Cc2d2a, Cep290, Prom1,* and *Ttc8* genes was not significantly affected unless all four *Msi1* and *Msi2* alleles were deleted (Fig 2). The photoreceptor-specific exon in the *Prom1* gene showed a minor 12% reduction in inclusion rates when only a single *Musashi* allele was expressed. The same effect on *Prom1* splicing was observed, regardless if the expressed allele was from the *Msi1* or *Msi2* gene (Fig 2). Complete deletion of all *Musashi* alleles resulted in significant downregulation of all four exons. These findings indicated that a single *Musashi* allele is sufficient to support high levels of inclusion of photoreceptor-specific exons.

### A single allele of *Musashi* is sufficient to maintain photoreceptor cell function

Previously, we have shown that the *Musashi* genes are required for photoreceptor development and for the maintenance of the mature photoreceptor cells. Deletion of both *Msi1* and *Msi2* in mature photoreceptors resulted in loss

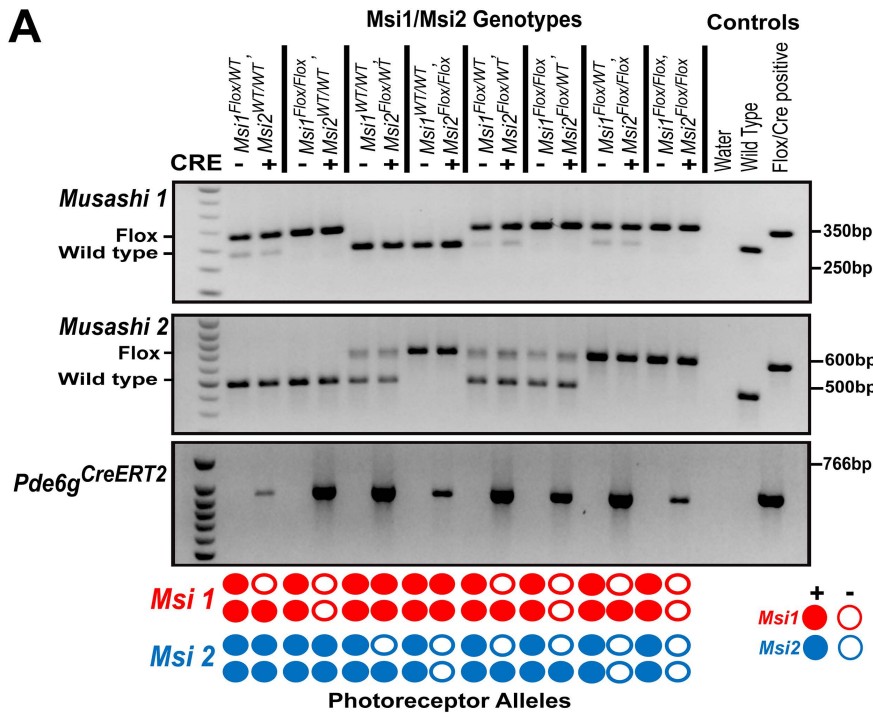

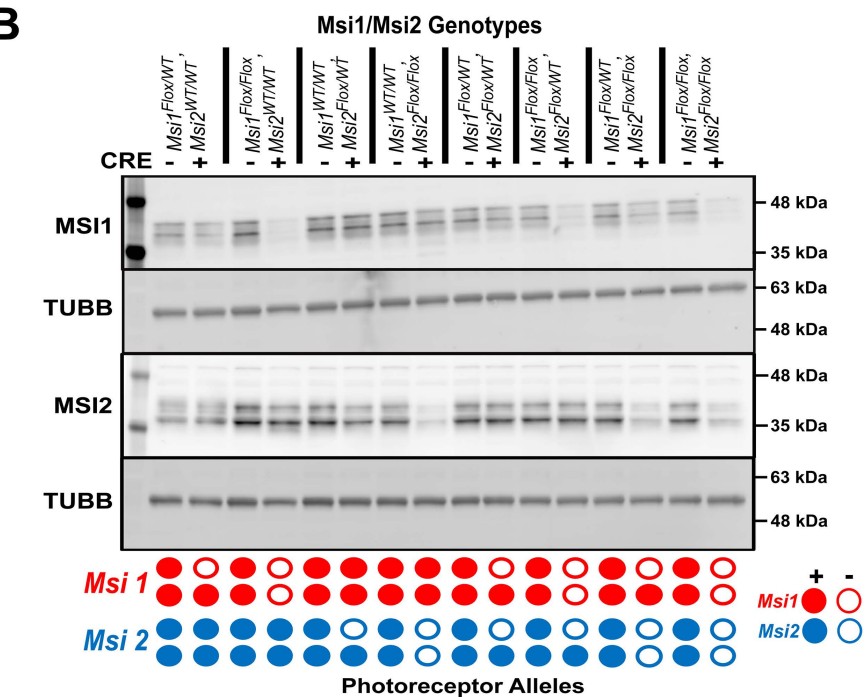

**Fig 1. Confirmation of allelic knockouts of *Musashi 1 (Msi1)* and/or *Musashi 2 (Msi2)*. (A)** Agarose gels showing the genotypes of the animals used in this study (only one replicate is shown here). Msi1/Msi2 genotypes are listed at the top. Below, "+" indicates Cre-positive and "-" indicates Cre-negative. Water (no template negative control), wild-type, and Flox/Cre-positive controls were used. "Flox" and "Wild type" denote *floxed* allele and *wild-type* allele, respectively. The Flox/Cre-positive control represents *Msi1*flox/flox, *Msi2*flox/flox, and *Pde6g*CreERT2. The diagram at the bottom indicates the knockout allele status in photoreceptor cells. Filled circles indicate the presence of the allele, while empty circles represent the absence of the allele. **(B)** Western blot showing the expression of MSI1, MSI2, and TUBB (as a loading control) from retinal tissues of different genotypes (labeled at the top). The retina were collected 14 days after the first tamoxifen injection at postnatal day 30.

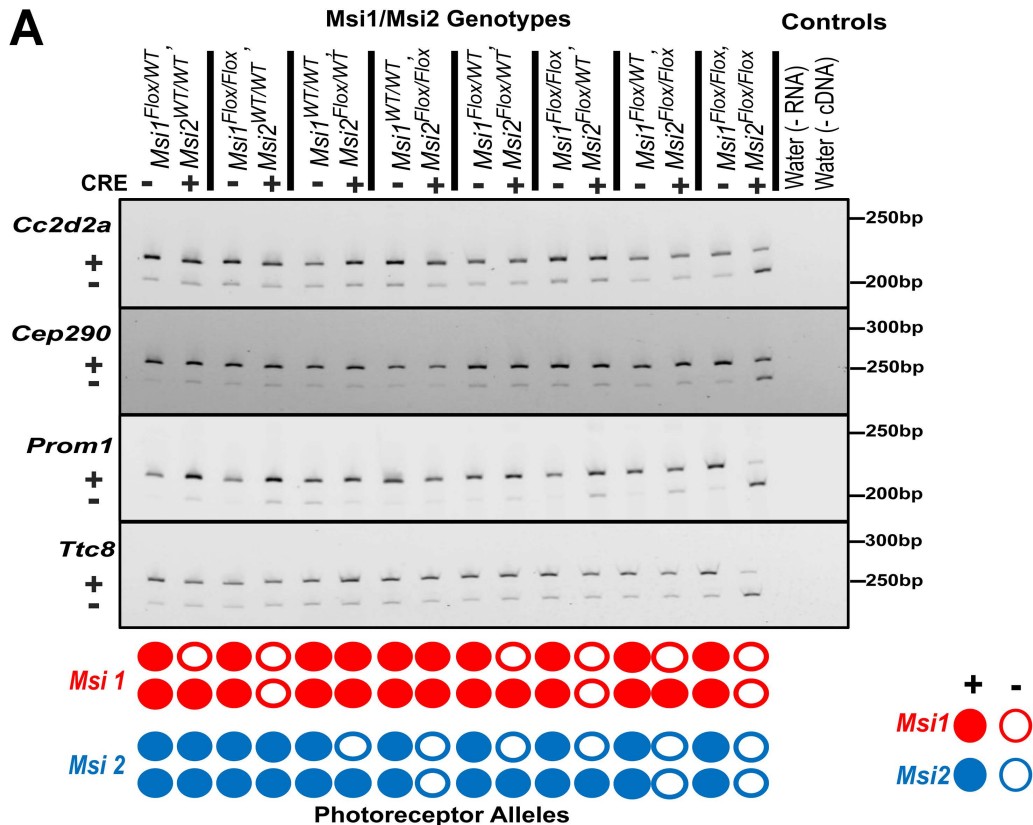

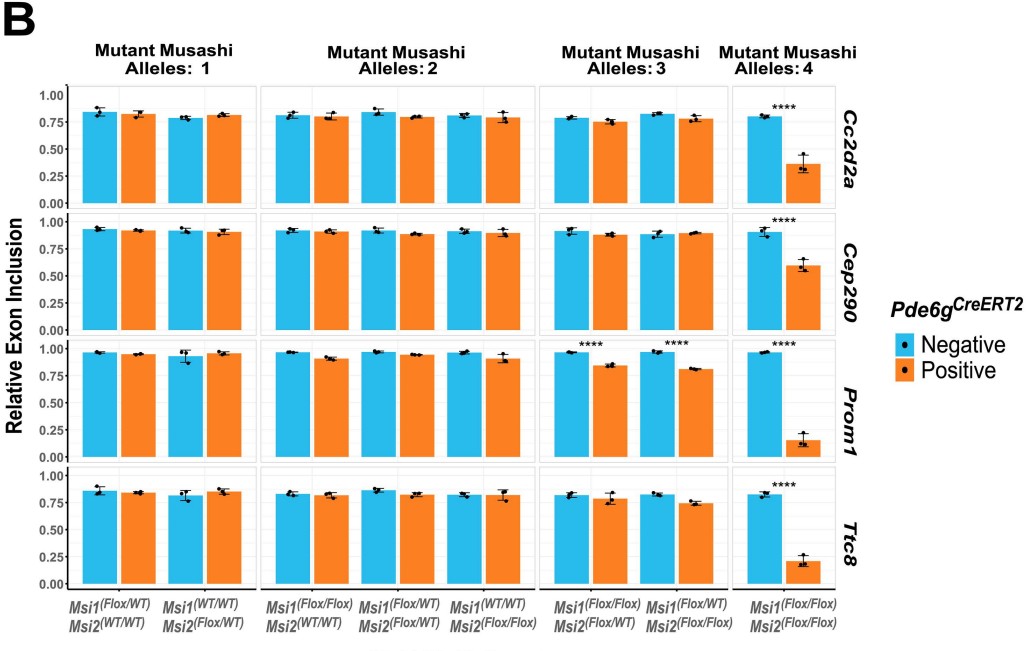

**Fig 2. Alternative exon splicing on photoreceptor-specific compound *Musashi* (*Msi*) allelic knockouts. (A)** RT-PCR analysis of four photoreceptor-specific alternative exons (*Cc2d2a exon 32*, *Cep290 exon 8*, *Prom1 exon 19*, and *Ttc8 exon 2a*) that were previously shown to be

dependent on MSI1 [21]. On the left, "+" indicates included exon and "-" indicates skipped exon.The retina were collected 14 days after the first tamoxifen injection at postnatal day 30. Genotypes for Msi1/Msi2/Cre are listed at the top of the gel images. The schematic at the bottom of each image indicates the status of the knockout alleles in photoreceptor cells. **(B)** Quantification of the exon inclusion levels, grouped by the number of *Musashi* alleles that were knocked out. Data are presented as mean±SEM. Statistical significance relative to the control was determined using Tukey HSD. Significance level is indicated as: *p-value < 0.05, **p-value < 0.01, ***p-value < 0.001.

of the retina response to light within 100 days after the knockout was induced [19]. As a single *Musashi* allele was sufficient to support splicing, we asked whether a single allele from either *Msi1* or *Msi2* could also support vision. We monitored the retina response to light by Electroretinogram (ERG) assessing both scotopic (rod photoreceptor) and photopic (cone photoreceptor) response to light in mice expressing a single *Msi1* or *Msi2* allele and matched controls (Fig 3 and S4 Fig). We did not observe a significant difference between the knockout animals and the controls for over 100 days after the knockout was induced. Both the A-wave corresponding to photoreceptor cell hyperpolarization and the B-wave corresponding to ON bipolar cell depolarization were normal, indicating that the phototransduction cascade and synaptic function in photoreceptor cells were unaffected. Thus, the physiological role for *Musashi* in vision can be fulfilled by a single *Msi1* or *Msi2* allele.

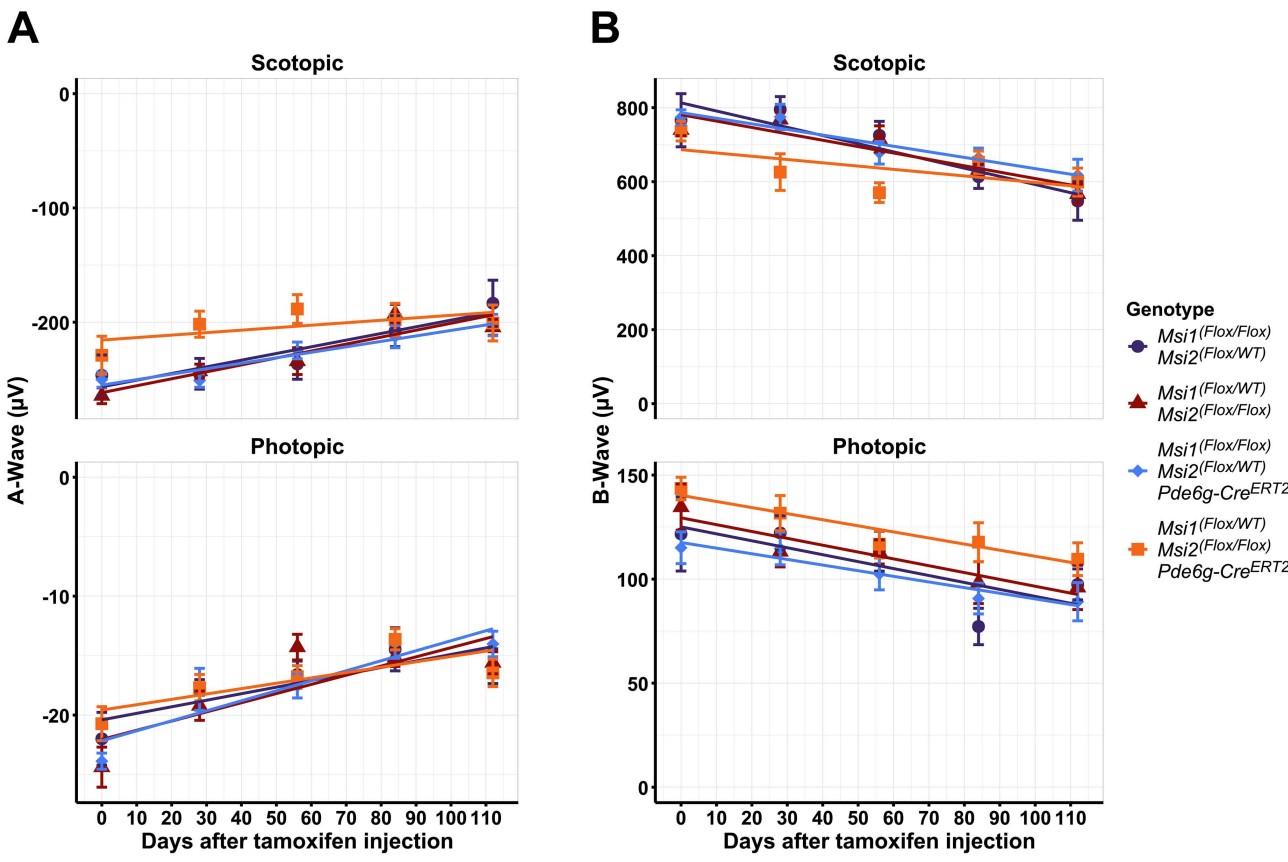

**Fig 3. Electroretinography (ERG) responses in photoreceptor-specific triple *Musashi (Msi)* knockouts.** Data were obtained from triple allelic knockouts from *Msi 1* and *Msi 2* (blue, orange) along with matched controls (purple and red). **(A)** Plots of A-wave amplitude from both scotopic (top) and photopic (bottom). **(B)** Plots of B-wave amplitude from both scotopic (top) and photopic (bottom). Scotopic ERGs were performed after overnight dark adaptation using −12 dB (0.151 cd*s/m²) flashes, and photopic ERGs were recorded after light adaptation using 3 dB (4.88 cd*s/m²) flashes.

## A feedback loop compensates for the loss of Musashi alleles

Neither alternative splicing nor light sensing were significantly affected in photoreceptors expressing a single *Msi1* or *Msi2* allele. Thus, we used immunofluorescence to determine if the *Msi1* and *Msi2* allele knockouts produced the expected decrease in protein levels and characterize the subcellular distribution of the two proteins. We probed sections from retinas expressing a single *Msi1* or *Msi2* allele with antibodies to MSI1 and MSI2. The immunofluorescence signal intensities in the photoreceptor inner segment and outer nuclear layer were quantified and normalized to the signal intensity of the inner neurons. We observed clear loss of MSI1 and MSI2 protein when both alleles for each gene were targeted (Fig 4A and S5-S8 Figs). However, we did

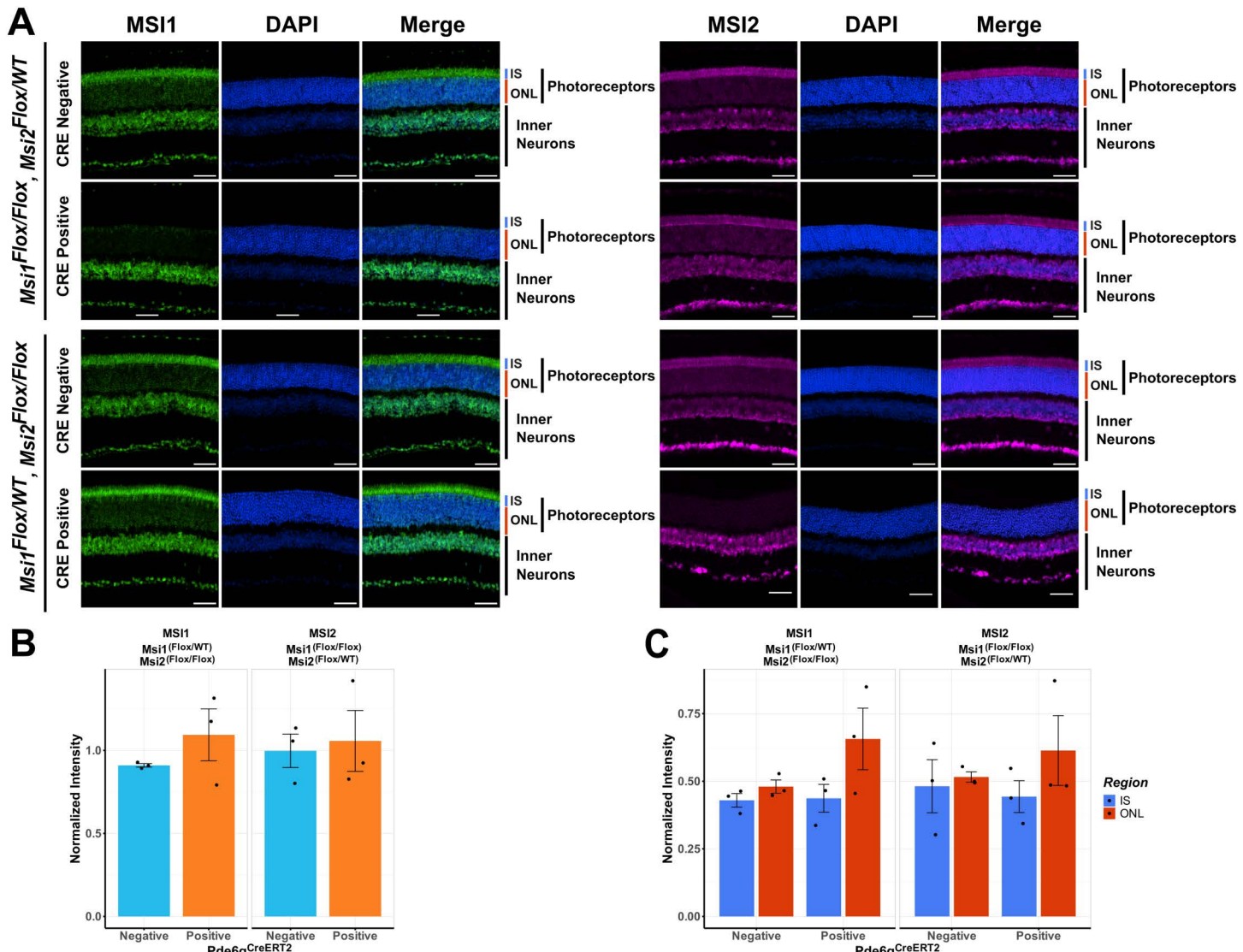

**Fig 4. Immunofluorescence analysis of photoreceptors expressing a single *Msi1* or *Msi2* allele.** Retinal sections were collected 143 days after the first tamoxifen injection from triple allelic knockouts and their genotype-matched controls. **(A)** Retinal cross-section stained against MSI1 (green), MSI2 (magenta), and DAPI for nuclei (blue) and scanned using laser lines, 488 nm (MSI1), 561 nm (MSI2), and 405 nm (DAPI) with a 40X objective. The scale bar is 50 μm. Retinal layers are labeled as IS for inner segment, ONL for outer nuclear layer, photoreceptors (IS + ONL), and inner neurons. **(B)** Quantification of MSI1 and MSI2 protein expression in photoreceptor cells in controls and animals expressing a single *Musashi* allele. The signal intensities are normalized to the signal in inner neurons where all *Musashi* alleles are intact. **(C)** Quantification of the MSI1 and MSI2 protein levels in the inner segment and outer nuclear layer of photoreceptor cells expressing a single *Musashi* allele and matched controls.

not observe a significant change in MSI1 or MSI2 protein levels in photoreceptors (the sum of the inner segment and outer nuclear layer signal) relative to the controls when either protein was expressed from a single allele of the respective gene (Fig 4A and 4B). This data indicates that a feedback mechanism exists that increases expression from the intact allele.

We also examined the distribution of the MSI1 and MSI2 proteins between the cytoplasm and the nuclei of the photoreceptor cells by comparing the immunofluorescence signal intensities in the inner segment (photoreceptor cytoplasms) and the outer nuclear layer (photoreceptor nuclei). Consistent with the preserved function in splicing, we did not observe a decrease in the MSI1 or MSI2 protein levels in the outer nuclear layer when only a single *Musashi* allele was expressed (Fig 4A and 4C). In fact, there was a trend for elevated *Musashi* protein levels in the outer nuclear layer in the knockout animals. However, this trend did not reach statistical significance.

Staining of with antibody to transducin alpha (GNAT1) and peanut agglutinin (PNA) revealed normal rod and cone photoreceptor outer segments in animals expressing a single *Msi1* or *Msi2* allele (Fig 5). Müller glia activation and gliosis are commonly observed in retinal injury and degeneration, including inherited retinal disease [25,26]. To assess Müller glia activation, we probed retinal sections from triple allelic knockouts and matching controls for glial fibrillary acidic protein (GFAP) (Fig 6 and S9 Fig). Astrocyte processes in the ganglion cell layer stained brightly for GFAP, providing an internal

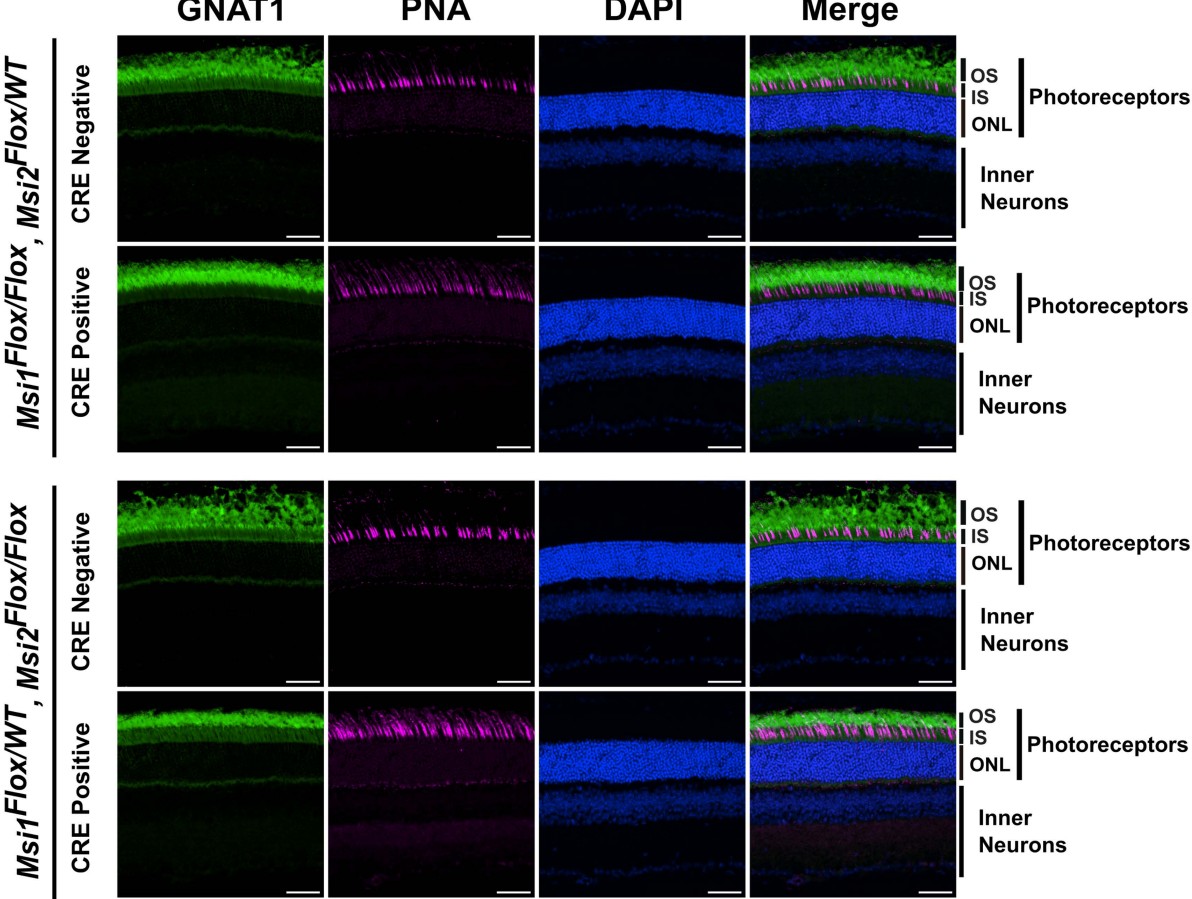

**Fig 5. Immunofluorescence analysis of retinal sections expressing photoreceptor markers, GNAT1 and PNA.** Retinal cross-section collected 143 days after the first tamoxifen injection from triple allelic knockouts and their genotype-matched controls, stained against GNAT1 (green), PNA (magenta), and DAPI for nuclei (blue) and imaged using laser lines, 488 nm (GNAT1), 561 nm (PNA), and 405 nm (DAPI) with a 40X objective. The scale bar is 50 µm. The labels are OS for outer segment, IS for inner segment, ONL for outer nuclear layer, photoreceptors (OS + IS + ONL), and inner neurons.

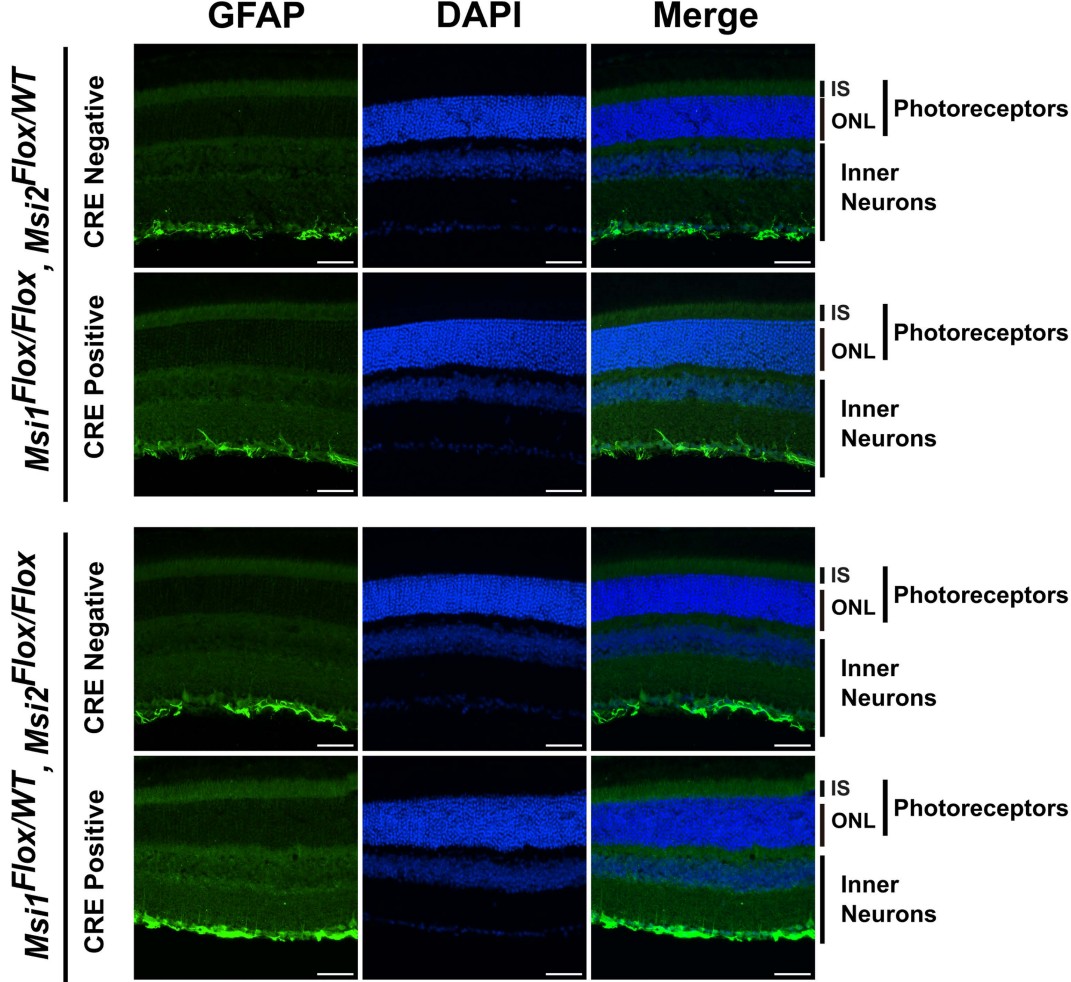

**Fig 6. Immunofluorescence analysis of retinal sections for GFAP.** Retinal cross-sections were stained for GFAP (green) and DAPI for nuclei (blue) and scanned using laser lines, 488 nm (GFAP) and 405 nm (DAPI) with a 40X objective. The samples were collected 143 days after the first tamoxifen injection from triple alleic knockouts and their genotype-matched controls. The scale bar is 50 µm. The labels are IS for inner segment, ONL for outer nuclear layer, photorecepotrs (OS + IS + ONL), and inner neurons.

positive control. We do not observe GFAP staining of Muller glia, indicating that Muller glia is not activated. The staining data shows normal photoreceptor outer segments and absence of inflammatory response.

## Discussion

*Msi1* and *Msi2* were partially redundant in developing photoreceptors, but were fully redundant in the mature photoreceptor [19,20]. This difference in the requirement for *Msi1* and *Msi2* was attributed to variations in the expression levels of the two proteins in developing photoreceptor cells. In the developing mouse retina, MSI1 protein levels sharply increase perinatally and peak one week after birth. In contrast, MSI2 protein levels were significantly upregulated after postnatal day 8 and reached its peak approximately thirty days after birth. Consequently, deletion of *Msi1* but not *Msi2* in retinal progenitors significantly reduced total *Musashi* protein levels, causing early vision defect and minor effect on alternative pre-mRNA splicing. The knockout of *Msi2* in retinal progenitors had little effect at the time of eye opening, but the response of

the retina to light progressively decreased with age. RNA binding by MSI1 is reported to interfere with nuclear import of the protein [22]. Thus, it seemed plausible that the photoreceptor-specific role of *Musashi* proteins in regulating alternative splicing can be achieved by increasing *Musashi* proteins levels so that they saturate their cytoplasmic targets. Indeed, the retina expresses exceptionally high levels of both MSI1 and MSI2 [20]. We used progressive reduction of *Musashi* gene dosage to test this hypothesis. We find that reducing gene dosage of *Msi1* and *Msi2*, short of complete knockout of all alleles from both genes, has little effect on splicing of photoreceptor-specific alternative exons. Similarly, the ability of photoreceptor cells to sense light remained unaffected when only a single *Msi1* or *Msi2* allele was present. The remarkable redundancy between *Msi1* and *Msi2* appears at least in part to be due to a robust feedback loop that upregulates protein expression from the intact *Musashi* allele in our knockout photoreceptors. Consequently, in our experiments, the *Musashi* protein levels do not decrease proportionally to the reduction in *Musashi* gene dosage. For this reason, we cannot conclusively rule out the stoichiometry between *Musashi* proteins and their cytoplasmic targets as a mechanism controlling the *Musashi* protein nuclear localization and function in pre-mRNA processing. Nevertheless, in the light of our results, such mechanisms appear less likely.

A limitation of the presented research is that the phenotype of the mice was tracked for a limited period of time. It is possible that subtle differences that have remained unnoticed may have revealed themselves if animals were tracked through their entire two year lifespan. Furthermore the splicing analysis is limited to four alternative exons. A more comprehensive RNA-Seq analysis can potentially reveal splicing events that show dose dependence across genotypes.

## Supporting information

**S1 Fig. Agarose gel electrophoresis of genotyping results for all three biological replicates used in Western blot and RT-PCR analyses.** Labels on the left indicate genes that were genotyped for. The first lane in each gel contains the molecular size standard in base pair (bp). Labels at the top and bottom denote the genotypes of the animals used in this study. "Flox" refers to the *floxed* allele. Red boxes represent the portion of the gels shown in Fig 1A.
(EPS)

**S2 Fig. Western blot images for all three biological replicates.** Labels on the left indicate the proteins that were probed for. Each blot includes a size standard marked in kDa. Labels at the top and bottom denote the genotypes of the animals from which the retina were collected for Western Blot analysis. Red boxes represent the regions of the blots shown in Fig 1B.
(EPS)

**S3 Fig. UREA-Polyacrylamide gel electrophoresis images from RT-PCR products for all three biological replicates.** Labels on the left indicate the genes whose alternative exon splicing was analyzed by RT-PCR. "+" and "-" indicate exons being included and skipped, respectively. Size standard is marked in base pair (bp). Labels at the top and bottom denote the genotypes of the animals obtained for the analysis. Red boxes highlight the portion of the gels shown in Fig 2A.
(EPS)

**S4 Fig. Electroretinography (ERG) plots.** Data were obtained from triple allelic knockouts from Msi 1 and Msi 2 (blue and orange) along with matched controls (purple and red). (A) Scotopic A-wave (top) and photopic B-wave (bottom) responses from the triple allelic knockout animals and the controls over time after the first tamoxifen injection. Scotopic ERGs were carried out after overnight dark adaptation using 0.151 cd*s/m2 flashes, and photopic ERGs were performed after light adaptation using 4.88 cd*s/m2 flashes. (B) The amplitude plot of A- and B-waves from triple allelic knockouts and genotype-matched controls over time after the first tamoxifen injection.
(EPS)

**S5 Fig. Portion of Immunofluorescence images of the retinal cross-sections for MSI1, cropped from Fig 4A.** Retinal sections were stained with antibody to MSI1 (green) and co-stained with DAPI (blue). Labels on the top indicate the stain. The genotype of the animals is indicated on the left of each group of images. Retinal layers are labeled as: IS (inner segment), ONL (outer nuclear layers), PR (photoreceptor cells, IS+ONL), and inner neurons. Scale bars are 50 μm. A 40X objective was used to collect images.
(EPS)

**S6 Fig. Portion of Immunofluorescence images of the retinal cross-sections for MSI2, cropped from Fig 4A.** Retinal sections were stained with antibody to MSI2 (magenta) and co-stained with DAPI (blue). Labels on the top indicate the stain. The genotype of the animals is indicated on the left of each group of images. Retinal layers are labeled as: IS (inner segment), ONL (outer nuclear layers), PR (photoreceptor cells, IS+ONL), and inner neurons. Scale bars are 50 μm. Images were acquired using a 40X objective.
(EPS)

**S7 Fig. Immunofluorescence images from retinal cross-sections for all three biological replicates for.** Retinal sections were stained with antibodies either to MSI1 (green) or MSI2 (magenta) and co-stained with DAPI (blue). Labels on the left indicate the stain. The genotype of the animals is indicated on the top of each group of images. Each column within each group is a separate biological replicate (different animal). Retinal layers are labeled as: IS (inner segment), ONL (outer nuclear layers), PR (photoreceptor cells, IS+ONL), and inner neurons. Scale bars are 50 μm. Red boxes frame the images shown in Fig 4A. A 40X objective was used to capture images.
(EPS)

**S8 Fig. Immunofluorescence images from retinal cross-sections for all three biological replicates.** Retinal sections were stained with antibodies either to MSI1 (green) or MSI2 (magenta) and co-stained with DAPI (blue). Labels on the left indicate the stain. The genotype of the animals is indicated on top of each group of images. Each column within each group is a separate biological replicate (different animal). Retinal layers are labeled as: IS (inner segment), ONL (outer nuclear layers), PR (photoreceptor cells, IS+ONL), and inner neurons. Scale bars are 50 μm. Red boxes frame the images shown in Fig 4A. Images were collected using a 40X objective.
(EPS)

**S9 Fig. Immunofluorescence images from retinal cross-sections for all three biological replicates for GFAP.** Retinal sections were stained with antibodies to GFAP (green) and co-stained with DAPI (blue). Labels on the left indicate the stain. The genotype of the animals is indicated on the top of each group of images. Each column within each group is a separate biological replicate (different animal). Retinal layers are labeled as: IS (inner segment), ONL (outer nuclear layers), PR (photoreceptor cells, IS+ONL), and inner neurons. Scale bars are 50 μm. Red boxes frame the images shown in Fig 6. All images were taken using a 40X objective.
(EPS)

**S1 File. Supplementary tables 1–6.** Tables 1 and 2 list reagents (antibodies, and primers). Tables 4–6 contain the data used to prepare the charts for Figs 2, 3, and 4.
(XLSX)

**S2 File. Immuno Fluorescence data analysis.**
(ZIP)

**S3 File. Raw ERG data and R-code.**
(ZIP)

## Acknowledgments

Mice carrying *Msi1^flox/flox* and *Msi2^flox/flox* alleles are a kind gift by Dr. Christopher Lengner (University of Pennsylvania).

## Author contributions

**Conceptualization:** Bohye Jeong, Peter Stoilov.

**Data curation:** Bohye Jeong, Peter Stoilov.

**Formal analysis:** Bohye Jeong, Peter Stoilov.

**Funding acquisition:** Peter Stoilov.

**Investigation:** Bohye Jeong, Peter Stoilov.

**Methodology:** Bohye Jeong, Peter Stoilov.

**Resources:** Peter Stoilov.

**Software:** Peter Stoilov.

**Supervision:** Peter Stoilov.

**Visualization:** Bohye Jeong, Peter Stoilov.

**Writing – original draft:** Bohye Jeong, Peter Stoilov.

**Writing – review & editing:** Bohye Jeong, Peter Stoilov.

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
