## [Decision Letter · Decision Letter 0]

13 Feb 2026

PONE-D-25-63099A single Musashi gene allele is sufficient to maintain mouse photoreceptor cellsPLOS One

Dear Dr. Stoilov,

Thank you for submitting your manuscript to PLOS ONE. After careful consideration, we feel that it has merit but does not fully meet PLOS ONE’s publication criteria as it currently stands. Therefore, we invite you to submit a revised version of the manuscript that addresses the points raised during the review process.

We look forward to receiving your revised manuscript.

Kind regards,

Gerrit Hilgen

Academic Editor

PLOS One

Journal Requirements:

2. To comply with PLOS One submissions requirements, in your Methods section, please provide additional information regarding the experiments involving animals and ensure you have included details on (1) methods of sacrifice, (2) methods of anesthesia and/or analgesia, and (3) efforts to alleviate suffering.

“R01EY025536”

5. We note that there is identifying data in the Supporting Information file “Supplementary_data_1_RawERG_data_and_R-code.zip” Due to the inclusion of these potentially identifying data, we have removed this file from your file inventory. Prior to sharing human research participant data, authors should consult with an ethics committee to ensure data are shared in accordance with participant consent and all applicable local laws.

-Location data

7. PLOS ONE now requires that authors provide the original uncropped and unadjusted images underlying all blot or gel results reported in a submission’s figures or Supporting Information files. This policy and the journal’s other requirements for blot/gel reporting and figure preparation are described in detail at https://journals.plos.org/plosone/s/figures#loc-blot-and-gel-reporting-requirements and https://journals.plos.org/plosone/s/figures#loc-preparing-figures-from-image-files. When you submit your revised manuscript, please ensure that your figures adhere fully to these guidelines and provide the original underlying images for all blot or gel data reported in your submission. See the following link for instructions on providing the original image data: https://journals.plos.org/plosone/s/figures#loc-original-images-for-blots-and-gels.

Additional Editor Comments:

Dear authors,

The reviewers would like to see a bit more extended discussion and more details about cell types and methodology. Most comments can be implemented on the existing data. However, I don't see a need to do more experiments.

Please see the review comments for details.

I am looking forward seeing your revised manuscript

Reviewer's Responses to Questions

**Comments to the Author**

1. Is the manuscript technically sound, and do the data support the conclusions?

Reviewer #1: Yes

Reviewer #2: Yes

2. Has the statistical analysis been performed appropriately and rigorously? 

Reviewer #1: Yes

Reviewer #2: Yes

3. Have the authors made all data underlying the findings in their manuscript fully available?

Reviewer #1: Yes

Reviewer #2: Yes

4. Is the manuscript presented in an intelligible fashion and written in standard English?

Reviewer #1: Yes

Reviewer #2: Yes

5. Review Comments to the Author

Reviewer #1: The manuscript by Jeong and Stoilov, "A Single Musashi Gene Allele Is Sufficient to Maintain Mouse Photoreceptor Cells," appears to be a continuation of the research being conducted by this group and is devoted to studying how many alleles of the Msi1 and Msi2 genes are required to maintain the normal structure and function of photoreceptors. The manuscript is written in good scientific English, the rationale for the research question is clearly presented, and the materials and methods are described accurately and thoroughly. The study was conducted at a high methodological level. The manuscript is sufficiently well illustrated to ensure the reliability of the results obtained. However, I have several recommendations/comments regarding the manuscript text that would be worth considering before it is published:

1) Since the authors recently published a paper on a very similar topic (The Musashi proteins MSI1 and MSI2 are required for photoreceptor morphogenesis and vision in mice, 2020), I would like to see a more detailed discussion in the introduction of what is new in the current manuscript, which is the reason for this clarifying work.

2) I would like to understand the basis for choosing the time points for performing different analyses after tamoxifen injections (e.g., 14 days for genotyping, 143 days for collecting retinas for immunohistochemistry). This should either be supported by some references, or you can cite your own experience.

3) Clarification is needed in the Materials and Methods section regarding what TUBB is and why it is used as a loading control. Some references are needed on where this approach was borrowed.

4) In the subsection 'A single allele of Musashi is sufficient to maintain photoreceptor cell function' in the Results section, was the ERG response examined to only one intensity? Figure 3 appears to show only one stimulus intensity. If so, how was this particular stimulus intensity chosen?

5) In this subsection, it is useful to emphasize that both photopic and scotopic ERGs were recorded, allowing us to differentiate between the preservation of rod and cone function.

6) Figure 3 – the current design is poor and unreadable. It has too many elements, and the animal group labels and axes are excessively small. It is necessary to consider the reader's needs and find a solution to make this figure more readable.

7) Caption to Figure 3, panels B and C (lines 262-263) – I strongly recommend replacing the word 'intensity' with 'amplitude'; this would be more terminologically correct.

Reviewer #2: In this manuscript, Jeong et Stoilov investigate the role of the Musashi genes, Msi1 and Msi2, in regulating photoreceptor-specific alternative splicing. By progressively reducing Musashi gene dosage in photoreceptors, the authors assess how allele number affects photoreceptor genes and photoreceptor function. Their results indicate that even a single Msi1 or Msi2 allele is sufficient to preserve splicing of photoreceptor-specific exons and maintain photoreceptor function.

The manuscript is clearly written, well-illustrated, and overall easy to follow. The study is interesting and the data are potentially valuable. However, several points should be clarified or addressed before this manuscript can be considered for publication.

Major comments:

• The authors should clearly indicate which retinal cell types express the Cre recombinase in the Pde6g-CreERT2 line. Is Cre activity restricted to rods, or present in both rods and cones?

• The authors state that Western blot confirmed “expected changes” in Msi1 and Msi2 levels. Please specify what those expected changes are. When only one allele is missing, it is difficult to appreciate whether the other allele leads to increased protein expression. This point should be developed.

• Among the genes tested for differential splicing, how many splice variants were covered by the primers? Could alternative splicing events occur due to imbalance of Msi1 / Msi2 levels? Did the authors also measure total gene expression levels, not only splice variants?

• Did the authors examine other splicing factors that might compensate via increased expression (e.g., PRPF family members)?

• Additional immunohistochemistry would strengthen the study, particularly for cilia and photoreceptor markers. GFAP staining would help assess retinal stress and Müller glia activation. Lack of GFAP upregulation would support absence of phenotype.

Minor comments:

• The text suggests a single intraperitoneal tamoxifen injection (p.5), but the Western blot section states that retinas were collected 14 days after the first tamoxifen injection. This implies that multiple injections may have been performed. Please clarify.

• Were splicing changes assessed at later time points? A compensatory effect could diminish over time, so longer follow-up would be informative.

• The schematic is helpful for readers. It might be easier to follow if placed between the MSI1 and MSI2 gels.

• Please verify whether the correct citation is Sundar et al., 2021 rather than Matalkah et al., 2022.

• One hundred days may be insufficient to detect subtle or slowly progressive defects. This limitation should be acknowledged.

• The legend states a 40× objective, but the images look closer to 20×. Please verify. In addition, a higher magnification of ONL/IS area in Fig. 4C should be added to better illustrate the findings.

• Can the authors comment on the strong increase of MSI1 or MSI2 in the INL when the other gene is completely missing?

• The authors are thanked for sharing their R scripts, but they cannot be fully tested because the file Light_intensity_calibration.csv is missing.

6. PLOS authors have the option to publish the peer review history of their article (what does this mean?). If published, this will include your full peer review and any attached files.

Reviewer #1: No

Reviewer #2: No

---

## [Author Response · Author response to Decision Letter 1]

6 Apr 2026

We would like to thank the reviewers for devoting time to read our manuscript and providing thorough and detailed critiques. Our responses to the specific issues are below.

Reviewer #1:

1) Since the authors recently published a paper on a very similar topic (The Musashi proteins MSI1 and MSI2 are required for photoreceptor morphogenesis and vision in mice, 2020), I would like to see a more detailed discussion in the introduction of what is new in the current manuscript, which is the reason for this clarifying work.

The rationale behind this study is not to demonstrate that the Musashi proteins are required for vision. As the reviewer points out we have already shown this (Sundar et al 2020, PMID: 33168629 and Matalkah et al 2022, PMID: 36153373). The rationale is related to the role of Musashi in regulating splicing specifically in photoreceptor cells, which is a nuclear function for the two proteins. It was previously shown that the nuclear localization signal of the Musashi proteins is on their RNA binding surface, which makes RNA binding and nuclear import mutually exclusive. In this paper we were testing the hypothesis that the Musashi proteins regulate splicing specifically in photoreceptor cells due to their high level of expression in this cell type. The logic was that the high levels of Musashi proteins saturate their cytoplasmic mRNA targets and the excess free protein can be imported into the nucleus to regulate splicing. We have edited the introduction (lines 58-60 in the track changes version of the revised manuscript) to make that point more clear.

2) I would like to understand the basis for choosing the time points for performing different analyses after tamoxifen injections (e.g., 14 days for genotyping, 143 days for collecting retinas for immunohistochemistry). This should either be supported by some references, or you can cite your own experience.

There must be some misunderstanding due to us combining the genotyping results and western blot analysis of the knockouts into one. We genotype the animals at weaning (post natal day 21). The knockout is induced at postnatal day 30 with injection of tamoxifen. The western blot analysis was performed 14 days later. The 14 day timepoint is based on our previous work (Matalkah et al 2022, PMID: 36153373), where we did a time course of Musashi protein levels and showed that two weeks after the knockout the protein levels reached their minimum. We amended the text (lines 233-236 in the track changes version of the revised manuscript) to clarify that point.

To reduce animal usage, eye cups for MSI1 and MSI2 immunofluorescence imaging (Figure 4) were from the animals that were used for ERG experiments. The ERG experiments were completed 112 days post tamoxifen injections and analysis took some additional time. This pushed the eye cup collection to 143 days post tamoxifen injection. We clarified this point on line 132 in the Materials and Methods section in the track changes version of the revised manuscript)

3) Clarification is needed in the Materials and Methods section regarding what TUBB is and why it is used as a loading control. Some references are needed on where this approach was borrowed.

TUBB refers to beta tubulin as recognized by the Sigma-Aldrich antibody T8328 (listed in Supplementary table 2). Beta tubulins are abundant cytoskeletal proteins in neural tissues. Beta tubulin antibodies are commonly used as loading controls, and we have used them as such in our previous studies of the Musashi proteins in mouse retina. While we could have used other loading controls or normalized to total protein, use of beta tubulin provides consistency with our previous research. We have amended the Materials and Methods section on lines 125-127 in the track changes version of the revised manuscript to clarify this point.

4) In the subsection 'A single allele of Musashi is sufficient to maintain photoreceptor cell function' in the Results section, was the ERG response examined to only one intensity? Figure 3 appears to show only one stimulus intensity. If so, how was this particular stimulus intensity chosen?

We routinely do our ERG experiments using four intensities ( -40 dB, -24 dB, -12 dB, -4 dB) to assess scotopic response, and 3 dB for photopic response. -12 dB intensity was chosen because it elicits strong rod photoreceptor response, without activating cones. We have added a supplementary figure 4B to show the scotopic intensity-response curves.

5) In this subsection, it is useful to emphasize that both photopic and scotopic ERGs were recorded, allowing us to differentiate between the preservation of rod and cone function.

We addressed this concern in lines 272-273 of the Results section in the track changes version of the revised manuscript.

6) Figure 3 – the current design is poor and unreadable. It has too many elements, and the animal group labels and axes are excessively small. It is necessary to consider the reader's needs and find a solution to make this figure more readable.

To make the figure 3 readable we moved panel A to supplementary figure 4A and enlarged the legend and labels text.

7) Caption to Figure 3, panels B and C (lines 262-263) – I strongly recommend replacing the word 'intensity' with 'amplitude'; this would be more terminologically correct.

We apologize for this error. Changed as suggested.

Reviewer #2:

Major comments:

• The authors should clearly indicate which retinal cell types express the Cre recombinase in the Pde6g-CreERT2 line. Is Cre activity restricted to rods, or present in both rods and cones?

While the Pde6g gene encodes the rod cGMP phosphodiesterase, it is expressed in cones and can compensate for the loss of the Pde6h gene that encodes the cone phosphodiesterase enzyme (Brennenstuhl et al 2015, PMID: 25739440). Consequently the Pde6g-CreERT2 knock-in line expresses Cre in both rod and cone photoreceptors. This is also evident from the immunofluorescence staining on Figure 4, where there are no cells in the photoreceptor layer that express MSI1 or MSI2 protein when both alleles of the respective gene are targeted. At the same time cones are readily detectable by PNA staining (Figure 5). We have previously shown (Matalkah et al 2022, PMID: 36153373) that deleting both MSI1 and MSI2 using Ped6g-CreERT2 abolishes both cone and rod photoreceptor response to light.

• The authors state that Western blot confirmed “expected changes” in Msi1 and Msi2 levels. Please specify what those expected changes are. When only one allele is missing, it is difficult to appreciate whether the other allele leads to increased protein expression. This point should be developed.

The expected change was 50-70% reduction of MSI1 or MSI2 protein in retinal lysates when both alleles of the respective genes are targeted in photoreceptor cells. As suggested we elaborated this point on lines 228-231 of the results section in the track changes version of the revised manuscript.

• Among the genes tested for differential splicing, how many splice variants were covered by the primers? Could alternative splicing events occur due to imbalance of Msi1 / Msi2 levels? Did the authors also measure total gene expression levels, not only splice variants?

Primers used for alternative splicing analysis on figure 2 are placed in the constitutive exons flanking a single alternative cassette exon. Each primer set covers the splicing of one alternative exon and detects two splice variants - one that includes and one that skips the exon. We have added “+” and “-” labels to the gel images in Figure 2 to mark the exon included and exon skipped variants. These splice variants should not be confused with the transcript variants produced by the gene - a gene can produce more than one transcript variant can include a particular alternative exon, depending on the splicing of other alternative exons in the same gene, and the use of alternative promoters and poly-adenylation sites.

We are not sure what exactly the reviewer means by “imbalance of Msi1 / Msi2 levels”. We can think of two interpretations - either different ratios of MSI1 to MSI2 protein, or differences in the combined expression of MSI1 and MSI2. We do not find a significant difference in the inclusion of the tested exons when only a single MSI1 allele is present compared to when only a single MSI2 allele is present, nor do we see any difference in splicing across genotypes that delete two alleles, regardless to which gene these alleles belong. This data would argue that the splicing is not affected by the ratio of MSI1 to MSI2. If we consider the combined MSI1 and MSI2 expression, loss of both proteins significantly impacts the splicing of the tested exons. In all cases the presence of a single Musashi allele is sufficient to promote the splicing of the tested exons to levels that were comparable to the controls.

We did not analyze total gene expression levels in the Musashi allele knockouts used in this work. We previously analyzed by RNA-Seq the transcript levels in the double Msi1/Msi2 knockout generated using identical Pde6G-CreERT2 induced knockout. In these experiments we did not observe significant changes in the transcript levels of Prom1, Cc2d2a, Cep290, and Ttc8 (Matalkah et al 2022, PMID: 36153373 - see Supplementary data 1 and 8 in that publication). Therefore we do not expect to see significant transcript changes in knockouts that preserve one or more of the Musashi alleles

• Did the authors examine other splicing factors that might compensate via increased expression (e.g., PRPF family members)?

We did not examine the levels of other splicing factors in this work, however we did that previously for the complete Msi1/Msi2 knockout in (Matalkah et al 2022, PMID: 36153373 - supplementary data 1). There were no significant changes in splicing factor transcript levels. Neither transcript nor protein levels of Prpf or other core spliceosome proteins were affected in the Msi1/Msi2 knockout. However, SRSF9 protein levels as detected by mass spec were significantly changed (1.7 fold increase adjusted p-value < 0.001; Matalkah et al 2022, PMID: 36153373 - supplementary data 5). The change in the protein levels of SRSF9 appears to be a result of increased skipping of its poison exon in the Musashi knockout.

• Additional immunohistochemistry would strengthen the study, particularly for cilia and photoreceptor markers. GFAP staining would help assess retinal stress and Müller glia activation. Lack of GFAP upregulation would support absence of phenotype.

We thank the reviewer for the suggestion. We performed GFAP staining on retinal sections from mice expressing single Msi1 or Msi2 allele (Figure 6). We do not see elevated GFAP expression in Müller glia from the retina of animals expressing a single Musashi allele. Astrocyte projections in the ganglion cell layer stained for GFAP and served as internal positive control.

Minor comments:

• The text suggests a single intraperitoneal tamoxifen injection (p.5), but the Western blot section states that retinas were collected 14 days after the first tamoxifen injection. This implies that multiple injections may have been performed. Please clarify.

Tamoxifen was injected on three consecutive days starting at postnatal day 30. Day count starts from the day of the first injection in the series. We apologize for the confusion - the materials and methods section for the tamoxifen injections was not clearly written. We have corrected the text of this section (lines 105-113 in the track changes version of the revised manuscript).

• Were splicing changes assessed at later time points? A compensatory effect could diminish over time, so longer follow-up would be informative.

We agree that compensatory effects can change over time. We have not assessed spicing changes at a later point. Due to the large number of genotypes we had to make a choice on how we allocate the samples for each experiment. Our choice was to use the longer time point samples for immunofluorescence. We have acknowledged the limited time course during which the phenotype of the animals was tracked as a limitation of the study (lines 406-409 in the track changes version of the revised manuscript).

• The schematic is helpful for readers. It might be easier to follow if placed between the MSI1 and MSI2 gels.

We tried to reformat Figure 1 as suggested, but this made the figure less readable for us as it spaced the gels apart. We prefer the keep in its current format.

• Please verify whether the correct citation is Sundar et al., 2021 rather than Matalkah et al., 2022.

We are not sure to which specific citation the reviewer refers to. After reviewing the citations we noticed one discrepancy where both Sundar and Matalkah were referenced regarding the inducible photoreceptor knockouts (line 223 in the track changes version of the revised manuscript) when only Matalkah was relevant. We have corrected this citation.

• One hundred days may be insufficient to detect subtle or slowly progressive defects. This limitation should be acknowledged.

We agree and we have acknowledged this limitation in the discussion (lines 406-409 in the track changes version of the revised manuscript).

• The legend states a 40× objective, but the images look closer to 20×. Please verify. In addition, a higher magnification of ONL/IS area in Fig. 4C should be added to better illustrate the findings.

We have used 40x objectives to image these sections. The image may appear as lower magnification because we used a larger field of view to show a broad section of the retina. We have added cutouts at larger magnification in supplementary figures 5 and 6.

• Can the authors comment on the strong increase of MSI1 or MSI2 in the INL when the other gene is completely missing?

This is likely due to variation of the staining across samples and replicates. This can be observed on supplementary figure 7 where the staining intensity of INL varies across replicates of the same genotype tracking the overall brightness of the image.

• The authors are thanked for sharing their R scripts, but they cannot be fully tested because the file Light_intensity_calibration.csv is missing.

We apologize for the omission. We have added the missing file to the archive.

---

## [Decision Letter · Decision Letter 1]

26 Apr 2026

A single Musashi gene allele is sufficient to maintain mouse photoreceptor cells

PONE-D-25-63099R1

Dear Dr. Stoilov,

We’re pleased to inform you that your manuscript has been judged scientifically suitable for publication and will be formally accepted for publication once it meets all outstanding technical requirements.

Kind regards,

Gerrit Hilgen

Academic Editor

PLOS One

Additional Editor Comments (optional):

Reviewers' comments:

Reviewer's Responses to Questions

**Comments to the Author**

1. If the authors have adequately addressed your comments raised in a previous round of review and you feel that this manuscript is now acceptable for publication, you may indicate that here to bypass the “Comments to the Author” section, enter your conflict of interest statement in the “Confidential to Editor” section, and submit your "Accept" recommendation.

Reviewer #1: All comments have been addressed

Reviewer #2: All comments have been addressed

2. Is the manuscript technically sound, and do the data support the conclusions?

Reviewer #1: Yes

Reviewer #2: Yes

3. Has the statistical analysis been performed appropriately and rigorously? 

Reviewer #1: Yes

Reviewer #2: Yes

4. Have the authors made all data underlying the findings in their manuscript fully available?

Reviewer #1: Yes

Reviewer #2: Yes

5. Is the manuscript presented in an intelligible fashion and written in standard English?

Reviewer #1: Yes

Reviewer #2: Yes

6. Review Comments to the Author

Reviewer #1: Even before the review, the work was well prepared and presented, and now the authors have adequately addressed the minor issues I pointed out. I am satisfied and believe the manuscript can be accepted for publication in its current form.

Reviewer #2: Most of my comments have been carefully addressed by the authors, and the manuscript has improved accordingly. The revisions have clarified several important points and strengthened the overall quality of the work. In its current form, I consider the manuscript suitable for publication.

7. PLOS authors have the option to publish the peer review history of their article (what does this mean?). If published, this will include your full peer review and any attached files.

Reviewer #1: No

Reviewer #2: No

---

## [Editor Report · Acceptance letter]

PONE-D-25-63099R1

PLOS One

Dear Dr. Stoilov,

I'm pleased to inform you that your manuscript has been deemed suitable for publication in PLOS One. Congratulations! Your manuscript is now being handed over to our production team.

Kind regards,

on behalf of

Dr. Gerrit Hilgen

Academic Editor

PLOS One